# Adherence to COVID-19 Preventive Measures in Mozambique: Two Consecutive Online Surveys

**DOI:** 10.3390/ijerph18031091

**Published:** 2021-01-26

**Authors:** António Júnior, Janeth Dula, Sérgio Mahumane, Olivier Koole, Sónia Enosse, Joseph Nelson Siewe Fodjo, Robert Colebunders

**Affiliations:** 1Direcção de Pesquisa em Saúde e Bem-Estar, Instituto Nacional de Saúde, 1120 Maputo, Mozambique; janet.dula@ins.gov.mz (J.D.); sergio.mahumane@ins.gov.mz (S.M.); sonia.enosse@ins.gov.mz (S.E.); 2International Center for AIDS Care and Treatment Programs (ICAP) at Columbia University, 1101 Maputo, Mozambique; ok2297@cumc.columbia.edu; 3Global Health Institute, University of Antwerp, 2000 Antwerp, Belgium; JosephNelson.SieweFodjo@uantwerpen.be (J.N.S.F.); Robert.colebunders@uantwerpen.be (R.C.)

**Keywords:** COVID-19, Mozambique, preventive measures, adherence, survey

## Abstract

We assessed adherence to government recommendations implemented shortly after the introduction of COVID-19 in Mozambique in March 2020, through two online cross-sectional surveys in April and June 2020. We quantified adherence to preventive measures by a composite score comprising of five measures: physical distancing, face mask use, hand hygiene, cough hygiene, and avoidance of touching the face. 3770 and 1115 persons participated in the first and second round respectively. Wearing face masks, regular handwashing and cough hygiene all reached compliance rates of over 90% while physical distancing and avoiding to touch the face reached a compliance rate of 80–90%. A multivariable model investigating factors associated with adherence found that being older, more educated, and belonging to the healthcare sector increased the odds for higher adherence. Private workers and retired people, respondents receiving COVID-19 information through social media, and those who reported flu-like symptoms were less likely to adhere. 6% of respondents reported flu-like symptoms which aligned with the WHO clinical definition of COVID-19, suggesting low level community transmission. In conclusion, most respondents in this online survey in Mozambique complied well with strategies to prevent COVID-19. Whether the good preventive behaviour explains the low grade COVID-19 transmission requires further study.

## 1. Introduction

On 11 March 2020 COVID-19 was declared a global pandemic by the World Health Organization (WHO, Geneva, Switzerland) [1]. In order to limit transmission, the WHO recommends minimizing contact between infected and non-infected persons, early detection and isolation of cases, and general personal and collective hygiene measures [2]. As part of these measures, the use of face masks, hand washing, physical distancing, cough etiquette and avoidance of crowded places persons are recommended.

The first COVID-19 case in Mozambique was detected on 22 March 2020, and one week later on 30 March, the President of Mozambique declared the State of Emergency, level 3 (with levels ranging from 1 to 4, and 4 being the most stringent level of social distancing) [3]. The order included closure of schools, restaurants and bars, churches and sports facilities, restricted gatherings to 10 people or less, restricted the number of people at the workplace, and restrictions on public transport and restrictions on movement. This state of emergency has been extended monthly (on 29 April, 28 May and 26 June) and was maintained up to 6 September 2020 [4].

By 29 November 2020, more than 15,000 confirmed COVID-19 cases had been diagnosed in Mozambique [5]. Maputo City is the epicenter of the COVID-19 epidemic in the country with about half of the diagnosed cases residing there; next comes Maputo Province accounting for about 17% of the diagnosed cases [5]. An earlier sero-prevalence survey showed a community prevalence of ~4% in Maputo City [6].

It is currently unclear how well people adhere to these measures over time and which factors determine adherence. We conducted this online survey to assess how well people in Mozambique respect the COVID-19 preventive strategies including social distancing, staying at home, and personal and collective hygiene measures, and assessed the factors associated with reduced adherence.

## 2. Methods

### 2.1. Study Setting and Design

We conducted two consecutive cross-sectional surveys in Mozambique (Figure 1): the first from 11 to 17 May and the second one from 9 to 22 June. This study was part of a network of online surveys organized by the International Citizen Project COVID-19 (ICPCovid; online platform available at: https://www.icpcovid.com/en/home) which uses web-based surveys to investigate the impact of COVID-19 and associated restrictions on residents of several low and middle-income countries. An international questionnaire to investigate the impact of COVID-19 and associated restrictions on residents of low and middle-income countries was developed by a team of twenty international ICPcovid consortium members, and has already been administered in several African countries as Uganda [7], the Democratic Republic of Congo [7], Somalia [8] and Nigeria [9]. This questionnaire was adapted by the research team in Mozambique and translated to Portuguese. We used a snowball approach for sharing the questionnaire through email and other social networks (like WhatsApp) while asking participants to further distribute the questionnaire with their contacts.

### 2.2. Data Collection

Using the online tool, we asked participants of 18 years and older for informed consent prior to data collection. We collected sociodemographic data, data on adherence to preventive measures like social distancing, staying at home, and personal and collective hygiene measures, and data on flu-like symptoms and having been tested for COVID-19.

Five-level Likert scores were used to assess fear or worry for oneself or others during the COVID-19 outbreak, and difficulties to adhere to the stay-at-home instructions; scores ranged from 1 (least level of fear/worry or difficulty) to 5 (highest level). All responses were submitted anonymously to the ICPcovid platform where they were stored in a password protected server in Belgium until data extraction.

### 2.3. Data Analysis

Descriptive statistics are presented using means with standard deviation (SD) for continuous outcomes, and percentages (%) for categorical variables. We used Pearson’s Chi-squared test to compare proportions across surveys and to investigate associations between two categorical variables. We used the Mann Whitney U test to compare continuous variables. The level of adherence to COVID-19 preventive measures was quantified by means of a composite score based on the responses to five questions (Table 1). An ordinal logistic regression model was constructed to investigate factors associated with adherence using the composite score as the dependent variable.

### 2.4. Ethical Considerations

The study protocol was approved by the Institutional Bioethics Committee of the National Institute of Health of Mozambique (Ref: 029/CIBS-INS/2020) and the University of Antwerp Ethics Committee.

## 3. Results

### 3.1. Participant Characteristics

We included 3770 and 1115 participants in the first and second round respectively (Table 2). About half of the participants came from an urban setting, ~35% from a sub-urban setting and ~15% from a rural setting. In the first and second survey, most respondents were male (57.7% and 58.6% respectively), had an undergraduate university level degree (68.9% and 70.5% respectively) and were government employees (33.4% and 46.1% respectively). More than half of the respondents worked from home on the day of the first and second survey (59.8% and 57.3% respectively). A small percentage were smokers (5.4% and 4.0% respectively) and had some chronic illness (21.4% and 20.5% respectively).

Overall, about 18% of respondents in both surveys reported experiencing at least one flu-like symptoms during the two weeks preceding the survey (17.5% during the first survey, and 18.4% during the second survey; *p* = 0.498). The proportion of participants meeting the WHO clinical definition of COVID-19 [10] was similar in both survey rounds (Table 2). Of the 69 respondents who reported to have been tested for COVID-19 during the second round, none was infected with the virus.

### 3.2. COVID-19 Preventive Behaviours

Wearing face masks, regular handwashing and cough hygiene all reached compliance rates of over 90%, physical distancing and avoiding to touch the face reached a compliance rate of 80–90%, and the regular use of alcohol-based gel was reported for about 65% of respondents (Table 3).

The mean adherence score of participants was similar during both surveys: 4.6 ± 0.7 (on a scale of 1 to 5). Upon combining data from the two surveys and investigating mean adherence scores by region, we found significant disparities (*p* = 0.001) with the highest adherence reported in the Gaza region, and the lowest adherence reported in Sofala (Table 4).

### 3.3. Analysis of Likert Scores

The mean level of worry/fear about respondents’ own health during the COVID-19 outbreak increased slightly from 2.7 ± 1.3 during the first survey to 2.8 ± 1.3 during the second survey (*p* = 0.021). Difficulty to stay at home remained stable during both surveys (3.1 ± 1.28 and 3.1 ± 1.29; *p* = 0.345). However, we found that it was more difficult to stay at home for men (*p* < 0.001) but found no difference in difficulty scores across residential setting (rural vs. sub-urban vs. urban; *p* = 0.110).

### 3.4. Factors Associated with Adherence to COVID-19 Preventive Measures

A multivariable model investigating factors associated with the adherence score to COVID-19 preventive measures found that being older, having attained secondary or undergraduate education levels, and belonging to the healthcare sector (as student or worker) increased the odds for higher adherence (Table 5). Private workers, retired people, those who received COVID-19 information from social media, and those who reported flu-like symptoms were less likely to adhere to the preventive measures.

## 4. Discussion

Our data show that most people in Mozambique who participated in the two surveys complied well with most strategies to prevent COVID-19 transmission. In particular, face mask use and regular hand washing were reported by more than 90% of the respondents. This high compliance with face mask use is in contrast with certain other African countries where face mask use is reported to be low (using similar research studies as the present study), such as in Uganda (32.7%) [7], Democratic Republic of Congo (43.2%) [7], Somalia (51.2%) [8] and Nigeria (64.5%) [9]. Few people reported to have been to a bar/restaurant or to have attended a religious gathering during the last seven days but more than half of the respondents had been to a market. As it is difficult to respect physical distance at markets, wearing face-masks should particularly be recommended at these places.

During both surveys a relatively large number of respondents (18%) reported experiencing at least one flu-like symptoms during the two weeks preceding the survey. In about 6% of all respondents, the flu symptoms met the criteria of the WHO clinical definition of suspected COVID-19 infection. However, all 69 participants (54 without a recent flu–like illness) with available COVID-19 results tested negative. This suggest low grade COVID-19 transmission in Mozambique during the period of the surveys. This has been confirmed by several provincial sero-surveys in Mozambique with community sero-prevalences ranging from less than 1% [11] to 5% [12].

Persons with low adherence were more likely to report a flu-like illness. This was contrary to our expectations; we anticipated that during a COVID-19 outbreak, the experience of flu-like symptoms would cause people to be more careful with their health and consequently improve their adherence to COVID-19 preventive measures. However, a more likely interpretation of this finding is that the flu-like symptoms were the consequences of low adherence to preventive measures.

Research has shown that face masks are an important tool to prevent the spread of respiratory pathogens [13]. However, the evidence that face masks also may protect a person from acquiring a respiratory infection is less well established [14].

A higher level of education, with the exception of postgraduate education level (only 7 respondents), was associated with better preventive behaviours. Participants who received COVID-19 information from social media were less likely to be adherent with the preventive measures suggesting that social media may have had a negative effect on the preventive behaviour of people. Therefore, it is important that governments disseminate clear, evidence based messages to the public about COVID-19 preventive measures. Being older was also related to high adherence to preventive measures. This is an important finding as older age is a risk factor for severe COVID-19 disease, and therefore the protection of the elderly is a priority during the current outbreak.

Despite the fact that most COVID-19 infection are reported from Maputo City, adherence with the COVID-19 preventive measures was relatively high in this province. The explanation for this may be that in this province which was the epicenter of the local outbreak, people considered themselves more at risk for COVID-19 infection and therefore may be more motivated to adhere to the preventive measures.

Some limitations of the study should be mentioned. To keep the questionnaire short and easier to fill, we assessed adherence to preventive measures using yes/no questions. Using Likert-like scale questions would have provided more precise information, albeit making the questionnaire lengthier. Similarly, the use of the term “regular” in quantifying the frequency of handwashing was not very precise and could be interpreted differently by different respondents. Moreover, self-reported responses may not reflect the real-life behaviour of participants.

People with no or limited internet access (less educated persons and those belonging to the lower social classes) were not able to participate in the research. People without internet connection generally will be older and are less educated but also may have been less exposed about fake news circulating on the internet. Therefore, our respondents cannot be considered as representative of the general population in Mozambique. As this was a cross-sectional survey, causality is difficult to assess, as we pointed out above when assessing the relationship between adhering to preventive measures and having flu-like symptoms.

## 5. Conclusions

In conclusion, these online surveys in Mozambique showed that most respondents complied well with most strategies to prevent COVID-19 transmission. In particular, adherence to face mask use and regular hand washing was high (more than 90%). However, a considerable number of participants attended markets where physical distancing can be challenging, which can be a risk factor for contracting COVID-19. Based on reported symptoms and data from local sero-surveys, there seems to have been low grade community COVID-19 transmission during the period of the surveys. Whether good COVID-19 preventive behaviour in Mozambique is responsible for this low grade of COVID-19 transmission requires further study. Complementing our study findings with qualitative research particularly among populations at higher risk for COVID-19 infection would certainly provide a better picture of the COVID-19 behavioural landscape in the country.

## Figures and Tables

**Figure 1 ijerph-18-01091-f001:**
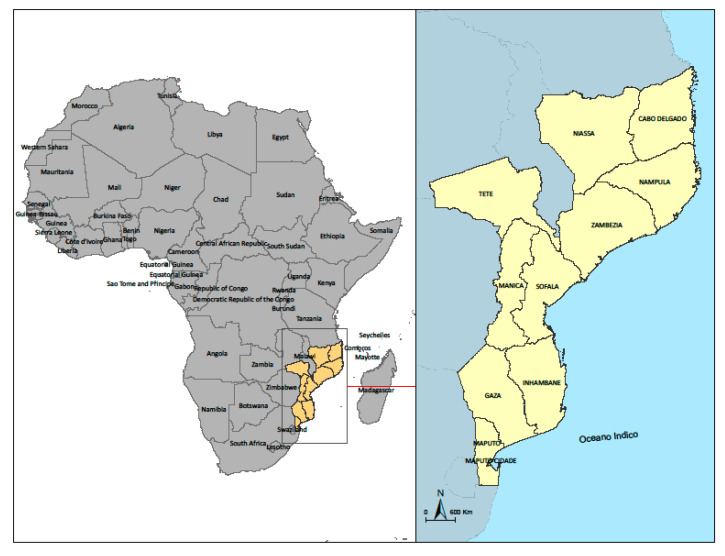
Map of Mozambique with its provinces.

**Table 1 ijerph-18-01091-t001:** Composition of the adherence score to COVID-19 preventive measures.

Variable	Scoring		Interpretation
I follow the 1.5–2 m physical distance rule	Yes	1	1 point for yes, 0 point for no
	No	0	
I wear a face mask when I go out	Yes	1	1 point for yes, 0 point for no
	No	0	
I wash hands regularly OR I use hand sanitizer	Yes	1	If any (or both) questions have answer yes, give just 1 point
	No	0	
When I cough/sneeze, I cover my mouth and nose	Yes	1	1 point for yes, 0 point for no
	No	0	
I avoid touching my face (eyes, nose, mouth)	Yes	1	1 point for yes, 0 point for no
	No	0	

Total adherence score (maximum): 5.

**Table 2 ijerph-18-01091-t002:** Participants’ characteristics during the two surveys in Mozambique.

Characteristics	Survey 1 Participants n = 3770	Survey 2 Participants n = 1115	*p*-Value *
Age: Mean (SD)	34.7 (10.6)	33.6 (9.2)	<0.001
Gender: n (%)			
Male	2174 (57.7%)	674 (58.6%)	0.594
Female	1596 (42.3%)	476 (41.4%)	
Education level: n (%)			
Primary school	17 (0.5%)	6 (0.5%)	0.704
Secondary school	1151 (30.5%)	332 (28.9%)	
University: Undergraduate	2596 (68.9%)	811 (70.5%)	
University: Postgraduate	6 (0.2%)	1 (0.1%)	
Residential setting: n (%)			
Rural	549 (14.6%)	187 (16.3%)	0.090
Sub-Urban	1353 (35.9%)	434 (37.7%)	
Urban	1868 (49.5%)	529 (46.0%)	
Live alone in household: n (%)	267 (7.1%)	88 (7.7%)	0.556
Profession: n (%)			
Private employee	1554 (41.2%)	355 (30.9%)	<0.001
Government employee	1261 (33.4%)	530 (46.1%)	
Student	497 (13.2%)	165 (14.3%)	
Unemployed	298 (7.9%)	70 (6.1%)	
Other	105 (2.8%)	22 (1.9%)	
Retired	55 (1.5%)	8 (0.7%)	
Working from home: n (%) ^a^	2043 (59.8%)	607 (57.3%)	0.166
Healthcare sector worker: n (%)	910 (24.1%)	446 (38.8%)	<0.001
Source of COVID-19 information: n (%)			
Social Media	453 (12.0%)	655 (57.0%)	<0.001
Smoking: n (%)	205 (5.4%)	46 (4.0%)	0.062
Presence of flu symptoms	660 (17.5%)	212 (18.4%)	0.498
Underlying chronic disease: n (%) ^b^	805 (21.4%)	236 (20.5%)	0.574
Tested for COVID-19: n (%)	NA	69 (6.0%)	NA
Positive COVID-19 test: n (%)	NA	0 (0.0%)	NA
Meet WHO clinical definition of suspected COVID-19 [7]: n (%)	239 (6.3%)	76 (6.6%)	0.797

NA: Not Applicable. * Chi squared test for categorical variables, Mann-Whitney U test for continuous variables. ^a^ Applicable only to workers. ^b^ Heart disease, diabetes, hypertension, cancer, HIV, or asthma.

**Table 3 ijerph-18-01091-t003:** COVID-19 preventive behaviours.

COVID-19 Preventive Behaviours	Survey 1 n = 3770	Survey 2 n = 1115	*p*-Value *
Individual measures			
Wear face masks: n (%)	3541 (93.9%)	1110 (96.5%)	0.001
Observe 1.5 m physical distancing: n (%)	3270 (86.7%)	945 (82.2%)	<0.001
Regular handwashing: n (%)	3636 (96.4%)	1097 (95.4%)	0.121
Regular use of alcohol-based gel: n (%)	2551 (67.7%)	769 (66.9%)	0.639
Cover mouth after coughing/sneezing: n (%)	3640 (96.6%)	1114 (96.9%)	0.668
Avoid to touch face (eyes, nose, mouth): n (%)	3094 (82.1%)	946 (82.3%)	0.917
Adherence score: n (%)			
0	3 (0.1%)	0 (0.0%)	0.042
1	13 (0.3%)	0 (0.0%)	
2	53 (1.4%)	12 (1.0%)	
3	246 (6.5%)	97 (8.4%)	
4	864 (22.9%)	276 (24.0%)	
5	2591 (68.7%)	765 (66.5%)	
Been to a bar/restaurant in the past 7 days: n (%)	264 (7.0%)	108 (9.4%)	0.009
Been to a market in the past 7 days: n (%)	2197 (58.3%)	736 (64.0%)	0.001
Been to a religious gathering in the past 7 days: n (%)	80 (2.1%)	25 (2.2%)	1.000
Meeting with more than 10 persons: n (%)	559 (14.8%)	248 (21.6%)	<0.001
Travelled during the past 7 days: n (%)	552 (14.6%)	194 (16.9%)	0.072

* Chi squared test.

**Table 4 ijerph-18-01091-t004:** Mean adherence scores by region in Mozambique during the study period.

Region	Number of Respondents, n (%)	Adherence Score, Mean (SD)
Cabo Delgado	98 (2.0%)	4.50 (0.74)
Gaza	333 (6.8%)	4.65 (0.68)
Inhambane	286 (5.8%)	4.62 (0.63)
Manica	140 (2.8%)	4.54 (0.78)
Maputo City	1331 (27.0%)	4.57 (0.69)
Maputo Province	1401 (28.5%)	4.61 (0.69)
Nampula	291 (5.9%)	4.49 (0.80)
Niassa	175 (3.6%)	4.62 (0.73)
Sofala	468 (9.5%)	4.47 (0.78)
Tete	170 (3.5%)	4.61 (0.73)
Zambezia	227 (4.6%)	4.60 (0.71)

**Table 5 ijerph-18-01091-t005:** Ordinal logistic regression model for Adherence score.

Covariate	Adjusted OR (95% CI)	*p*-Value
Age	1.03 (1.02–1.04)	<0.001
Male gender	1.04 (0.92–1.18)	0.551
Profession		
Government employee	Ref	
Private employee	0.86 (0.74–0.99)	0.041
Student	0.90 (0.72–1.11)	0.326
Other	0.97 (0.65–1.46)	0.872
Unemployed	1.09 (0.84–1.41)	0.532
Retired	0.43 (0.24–0.78)	0.004
Residential setting		
Rural	Ref	
Sub-urban	0.99 (0.82–1.20)	0.937
Urban	1.03 (0.86–1.23)	0.772
Educational level		
Primary	Ref	
Secondary	3.25 (1.38–7.52)	0.006
Undergraduate	2.93 (1.24–6.78)	0.012
Postgraduate	4.26 (0.77–33.81)	0.117
Healthcare sector worker	1.17 (1.01–1.36)	0.032
Living alone in household	0.90 (0.72–1.14)	0.369
Presence of flu symptoms	0.64 (0.55–0.74)	<0.001
COVID-19 information from social media	0.72 (0.61–0.84)	<0.001
Survey round		
Round 1	Ref	
Round 2	1.04 (0.89–1.23)	0.599

OR: Odds ratio; CI: Confidence interval; Ref: Reference category.

## Data Availability

The data are available upon reasonable request through the ICPcovid website: https://www.icpcovid.com/.

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
