# Peer review of "Adherence to COVID-19 Preventive Measures in Mozambique: Two Consecutive Online Surveys"

_ijerph, 2021, doi:10.3390/ijerph18031091_

Round 1
Reviewer 1 Report
The authors have adequately addressed prior review comments.
Reviewer 2 Report
the reported deficiencies have been adjusted. because there is no validity of the list, it remains difficult to translate the scores into the actual situation
This manuscript is a resubmission of an earlier submission. The following is a list of the peer review reports and author responses from that submission.
Round 1
Reviewer 1 Report
This is an interesting article about adherence to COVID-19 prevention recommendations in Mozambique that is highly relevant to this scientific moment; however, more detail is needed regarding the methodology, results, and limitations.
The methodology section would benefit from more clarity about the International Citizen Project COVID-19 platform. IRBs in both Mozambique and Belgium are cited as having reviewed the protocol, but what were the roles of investigators at each location? Did the Mozambique-based research team develop the questions, for example, or did they select them from a menu developed for the platform as a whole? Currently, sharing COVID-19 survey interests across institutions is not uncommon. For example, in the U.S., the National Institutes of Health have an online repository of COVID-19 research tools. Thus, it would be helpful to understand the roles and responsibilities of investigators in this collaboration.
The selection of the adherence questions needs some discussion. As I read the manuscript, respondents were provided a binary yes/no choice about adherence to specific recommendations. It would be helpful to know why that approach was selected rather than a Likert-like scale of responses that could allow respondents to select on a scale from Always to Never, which might provide additional insight into adherence.
More information about recruitment also would be helpful. Some readers may be unfamiliar with the WhatsApp platform, so additional background would be useful, particularly since the recruitment mechanism lends itself to questions about sample bias.
In the Results section, Table 4 requires additional information. What are the distinguishing demographic/geographic/economic characteristics of these provinces that could be driving adherence levels? Consider adding those as columns to the table. One suggestion: do statistics exist regarding the percentage of government versus private workers in each district? I ask because the table makes it look as if Maputo Province reported some of the highest levels of adherence to recommendations; however, elsewhere in the manuscript you note that Maputo Province was responsible for approximately half of the diagnosed cases. Could this apparent contradiction result from sampling bias (i.e., primarily government workers were sampled, but do they comprise a smaller proportion of that province's population)?
In the Discussion, please report the sources for mask adherence numbers across countries so that the reader can determine whether an apples-to-apples comparison is being made. Also in the discussion, it seems highly likely to me based on the timing of the survey and assessment of the symptoms that the directionality for the finding described in paragraph two is "low adherence likely contributes to more symptoms." In other words, I think your paragraph probably discusses this correlation in the wrong order.
Thank you for the opportunity to read your interesting research.
Author Response
Dear Reviewer,
please find our responses in the attached document.
Best
Olivier

Reviewer 2 Report
1 Self-observation often does not correspond to everyday reality
2 Regular is a term that cannot be quantified. How often ........is quantifiable
3 I miss research into the use of the correct coughing technique when coughing
4 I miss an estimate of what the bias is by having or not having internet access in the results
Reviewer 3 Report
This work by Junior et al., has investigated the adherence to COVID-19 preventive measures in Mozambique with two consecutive online surveys.
COVID-19 affects to all over the world, and it is an urgent matter to prevent the spread with the daily life behavior until the effective vaccine and/or antivirals will be developed. Many studied have shown that minimizing contact between infected and non-infected person, using face masks, washing hands, cover a mouth and nose when cough/sneeze, avoid touching face, could limit the virus spread.
Based on the above background, authors survey the adherence of those preventive measures in Mozambique in two time points (April and June). The study was well designed and analyzed well. Discussion was also described well enough to interpretate the data, including the description of the limitation of the study. However, there is some points which the reviewer recommended to add to easily understand for the readers as well as to understand their data objectively.
First point was to ask adding the information of the general case number of the flu-like symptoms in Mozambique at this term (April-June). Is there any significant difference of the case number on flu-like symptoms between this year (2020) and the past years (-2019)?
Second point was to add the geographical map to interpretate more accurately of the city/province where authors showed in Table 4 and also described in the introduction (Maputo Province).
Third and the last comment was is there any similar study/survey reported from the close country or in Africa? If so, please add some discussion based on that findings.
Overall, the manuscript was well designed and written, but few more information would assist readers to understand easier of the study.
Round 2
Reviewer 1 Report
Thank you for the careful attention paid to all reviewers' comments in your manuscript. You might want to do one last check for consistent capitalization (e.g., Maputo city vs. Maputo City) and any spelling or grammatical issues.
Reviewer 2 Report
I am missing the validation data of the questionnaire that was used. This is necessary to find out what the relationship is between the data found and the actual situation
further
1 Self-observation often does not correspond to everyday reality
2 Regular is a term that cannot be quantified. How often ........is quantifiable
3 I miss research into the use of the correct coughing technique when coughing
4 I miss an estimate of what the bias is by having or not having internet access in the results